# Bacille Calmette-Guérin Site Reactivation of Kawasaki Disease in Infants under 3 Months of Age: Relation with Diagnosis and Prognosis

**DOI:** 10.3390/children9060857

**Published:** 2022-06-08

**Authors:** Da Eun Roh, Jung Eun Kwon, Yeo Hyang Kim

**Affiliations:** 1Division of Pediatric Cardiology, Kyungpook National University Children’s Hospital, Daegu 41404, Korea; ponyks1004@gmail.com (D.E.R.); lovecello623@gmail.com (J.E.K.); 2Department of Pediatrics, Busan Paik Hospital, Inje University College of Medicine, Busan 47392, Korea; 3Department of Pediatrics, School of Medicine, Kyungpook National University, Daegu 41944, Korea

**Keywords:** infants, mucocutaneous lymph node syndrome, Bacille Calmette-Guérin

## Abstract

Diagnosis of Kawasaki disease in infants under 3 months of age is challenging. This study aimed to confirm the diagnostic efficacy of BCGitis in patients with Kawasaki disease aged <3 months. Overall, 473 children were enrolled; they were grouped by age into group 1 (≤3 months, *n* = 19) and group 2 (>3 months, *n* = 454). Data, including clinical features and laboratory results, were analyzed and compared between the groups. In group 1, 89% of patients showed Bacille Calmette-Guérin site reactivation. In group 1, total duration of fever and fever duration before initial treatment were significantly shorter than in group 2 (*p* = 0.001). The incidences of conjunctival injection, changes in extremities (erythema and edema), and cervical lymphadenopathy were significantly lower (*p* = 0.006, *p* = 0.040, and *p* < 0.001, respectively), and desquamation was higher in group 1 (*p* = 0.004). The incidences of incomplete Kawasaki disease, coronary artery complications, and resistance to intravenous immunoglobulin did not differ between the groups. Kawasaki disease should be suspected in infants aged <3 months with unexplained fever and BCGitis, even if the principal clinical symptoms are not fully presented. BCGitis in infantile Kawasaki disease is a useful sign and can help in the diagnosis of Kawasaki disease.

## 1. Introduction

Kawasaki disease most commonly occurs in children under five years of age. The incidence peaks between six months and two years of age [1,2]. The incidence of Kawasaki disease in infants under three months of age is lower than in older children, and reports of neonatal Kawasaki disease are extremely rare. The incidence of infantile Kawasaki disease in infants under three months of age in Korea and Japan is only 2.2% and 1.7%, respectively [3,4]. Diagnosis of Kawasaki disease in young infants under three months of age is challenging because only a few cases meet four of the five classical clinical criteria. Consequently, increased risk of delayed diagnosis and treatment can lead to increased cardiac complications.

In regions of the world where tuberculosis immunization with Bacille Calmette-Guérin (BCG) vaccine is still administered, more than 70% of patients with complete Kawasaki disease between six and 20 months of age showed inflammation at the BCG inoculation site (BCGitis) [5]. Previous studies have confirmed the importance of BCGitis as an early predictor of infantile Kawasaki disease. Based on this, BCGitis was included in the principal criteria of the sixth revision of the Japanese diagnostic guidelines for Kawasaki disease [2,6].

The purpose of this study was to investigate the diagnostic usefulness of BCGitis in very young patients aged less than three months with Kawasaki disease.

## 2. Materials and Methods

### 2.1. Study Participants and Data Collection

We retrospectively reviewed the medical records of 473 patients with Kawasaki disease who were admitted to Kyungpook National University Children’s Hospital in South Korea between January 2016 and April 2020. Patients were grouped by age into group 1 (≤3 months, *n* = 19) and group 2 (>3 months, *n* = 454). The following demographic variables were analyzed from a review of medical records: the median age, sex, fever duration, total duration of hospitalization, and the number of clinical symptoms that met the principal criteria.

Patients with fever for more than five days and at least four of the five principal criteria (i.e., lip redness, strawberry tongue, bilateral conjunctival injection, cervical lymphadenopathy, changes in extremities, such as erythema and edema of the palms and soles, and rash, including redness at the BCG inoculation site) were diagnosed with complete Kawasaki disease [7]. Incomplete Kawasaki disease was diagnosed when patients showed less than four of the principal criteria. If BCGitis appeared in a febrile patient whose cause of fever was not clear, Kawasaki disease was considered as a possible diagnosis, even if other symptoms were lacking. In South Korea, BCG vaccination is administered through the transdermal or intradermal route, and the manifestations of BCGitis may differ depending on the route of vaccination (Figure 1). Refractory Kawasaki disease is diagnosed when fever is present for at least 36 h after the end of initial treatment with intravenous immunoglobulin (IVIG).

### 2.2. Laboratory Test

Laboratory tests were performed on the first day of admission. The measured parameters were white blood cell (WBC) count, neutrophils, hemoglobin, platelet counts, C-reactive protein (CRP), erythrocyte sedimentation rate (ESR), aspartate aminotransferase (AST), alanine aminotransferase (ALT), N-terminal pro-brain natriuretic peptide (NT-ProBNP), and serum total protein albumin levels.

### 2.3. Echocardiography

Echocardiography was performed in all patients to evaluate cardiovascular complications in the acute phase (within one week of hospitalization) and at the end of the convalescence phase (eight weeks after the onset of illness) in all patients. We calculated the z-score of coronary artery diameter based on the diameter of the aortic valve annulus [8]. The luminal diameters of the left main coronary artery and the proximal right coronary artery were measured from parasternal short-axis views. The luminal diameter of the aortic valve annulus was assessed from parasternal long-axis views. Coronary artery dilatation was defined when the z-score of the coronary artery size was greater than 2.0. Coronary artery aneurysm was defined as a z-score of the coronary artery size greater than 2.5 [7].

### 2.4. Statistical Analysis

All statistical analyses were performed using IBM SPSS Statistics for Windows (version 26.0; IBM Co., Armonk, NY, USA). Chi-square and independent *t*-tests were performed to compare clinical characteristics and laboratory data. Statistical significance was set at *p* < 0.05.

### 2.5. Ethics Statement

This study was reviewed and approved by the institutional review board of Kyungpook National University Chilgok Hospital, approval number KNUCH 2021-11-012 (approval date: 22 November 2021). Written informed consent has been obtained from the patient’s parents to publish this paper.

## 3. Results

### 3.1. Clinical Characteristics

The demographic data and clinical characteristics of the two groups are summarized in Table 1.

Among the 473 patients included, 19 (4%) were under three months of age (group 1). The median age of patients in group 1 was 90 (82.5–99.5) days and none were younger than 60 days.

There was no significant difference in the proportion of patients diagnosed with complete versus incomplete Kawasaki disease. The incidence of BCGitis in group 1 was higher than that in group 2 (*p* < 0.001). Of all patients in both groups, 73% of BCGitis patients were less than 18 months and 11% were less than three months.

The incidences of conjunctival injection (*p* < 0.001), cervical lymphadenopathy (*p* < 0.001) and erythema and edema of the extremities (*p* = 0.040) were significantly lower in group 1 than in group 2. Desquamation of the fingers and toes (*p* = 0.004) was significantly higher in group 1. The incidences of lip redness/fissure, strawberry tongue, and skin rash did not differ significantly between the groups.

### 3.2. Laboratory Findings

The results of laboratory tests of the two groups are summarized in Table 2.

CRP, ESR, and total protein levels were significantly lower in group 1 than in group 2 (*p* = 0.049, *p* = 0.002, and *p* < 0.001, respectively) on the first day of admission. The platelet count and NT-proBNP level were significantly higher in group 1 than in group 2 (*p* = 0.031 and *p* = 0.034, respectively). WBC count and AST, ALT, and total bilirubin levels were not significantly different between the groups.

### 3.3. Clinical Course

In group 1, the total duration of fever (4 (IQR 3–4.5) vs. 6 (IQR 5–7); *p* < 0.001) and duration of fever before IVIG administration (2 (IQR 2–3) vs. 5 (IQR 4–6); *p* < 0.001) were significantly shorter than in group 2.

In group 1, all patients received IVIG (2 g/kg) as the first treatment, and four of them showed resistance to IVIG treatment. Of the four IVIG-resistant patients, one patient received repeated IVIG (2 g/kg) treatment, and three patients received high-dose corticosteroid (intravenous methylprednisolone, 30 mg/kg/day). Of the patients treated with high-dose corticosteroids, two had complete relief of fever; however, one patient became resistant to second-line treatment and received intravenous infliximab (5 mg/kg) as an additional treatment. The occurrence of IVIG resistance was not significantly different between the groups.

### 3.4. Cardiovascular Complication

The occurrence of cardiovascular complications did not differ between the groups. In group 1, no patient showed coronary artery lesions, but two patients (11%) had coronary artery aneurysms in the acute phase, and they showed regression to normal luminal dimension at the two-month follow-up. In group 2, 16 patients had coronary artery lesions and 29 patients had coronary artery aneurysms. Among them, 33 patients showed regression of coronary artery abnormalities and 12 patients showed persistent coronary artery abnormalities at the one-year follow-up.

In this study, there was no patient with systolic ventricular dysfunction, myocardial hyperechogenicity or pericardial effusion in echocardiography. There was no patient with abnormal electrocardiogram results, such as arrhythmias or ST-segment changes.

## 4. Discussion

The present study showed that when BCGitis was considered as an early symptom of Kawasaki disease in young infants: (1) the number of Kawasaki disease patients with BCGitis was significantly higher in patients younger than three months than in older children; (2) Kawasaki disease was diagnosed and treated earlier, resulting in a shorter duration of fever; and (3) there was a low risk of treatment resistance or cardiac complications because diagnosis and treatment could be made earlier.

Kawasaki disease occurs in children under five years of age, particularly between six and 20 months of age [1,2]. The incidence of Kawasaki disease is reported to be low in infants under six months of age. The incidence of Kawasaki disease is markedly low in the first two months of life [9], and the incidence of Kawasaki disease under three months of age is 2.2% [4]. In this study, the incidence of Kawasaki disease in patients aged less than three months was 4%, which is similar to that reported in previous studies.

The diagnosis of Kawasaki disease in young infants is difficult because it is rare and seldom meets four of the five principal criteria. This delays the initiation of treatment and is consequently associated with an increased risk of coronary artery complications. Therefore, a high index of suspicion is required during the diagnostic evaluation of young infants with fever. In previous studies, BCGitis has been reported as a useful diagnostic clue for suspected incomplete Kawasaki disease [10,11,12].

BCGitis occurs in the early stages of Kawasaki disease. Patients with complete Kawasaki disease with BCGitis have been shown to visit the hospital on average between one and four days after onset [5,10,12,13,14]. Therefore, BCGitis can lead parents or guardians to visit the hospital quickly if the patient has a fever or other symptoms of Kawasaki disease.

A Japanese nationwide epidemiological survey found that >70% of patients aged between three and 20 months with complete Kawasaki disease had BCGitis [5]. Looking at the age distribution of Kawasaki disease with BCGitis, the incidence peaked at six months of age and then decreased with age [5,10,11]. In Korea, more than 70% of Kawasaki disease cases with BCGitis are <18 months of age, and the peak incidence is observed at 6–8 months and then decreases [15]. In the present study, 34% of all patients had BCGitis, and the median age was 10 (IQR 6–19) months.

The occurrence of BCGitis was significantly higher in Kawasaki disease patients than in other acute febrile illnesses [16]. Compared to non-Kawasaki disease febrile illness, BCGitis occurs specifically in Kawasaki disease and is a significant predictor of Kawasaki disease with a high positive predictive value and a low negative predictive value [5]. Extensive edema and marked capillary dilatation in the papillary dermis were observed on the skin biopsies of the BCG inoculation site of Kawasaki disease patients. In these skin lesions, cytokine levels, such as IL-1α and TNF-α, were increased [17]. The reason for BCGitis in Kawasaki disease is thought to be a cross-reaction of the human homolog heat shock protein (HSP) 63 with mycobacterial HSP 65 [10,18]. The presence of a bull’s eye pattern on dermatoscopy at the BCG inoculation site is associated with coronary artery complications in Kawasaki disease [19]. BCGitis was not related to ethnicity, sex, or the number of Kawasaki disease features.

BCGitis has become one of the diagnostic criteria included in the Japanese Circulation Society/Japanese Society for Cardiovascular Surgery 2020 and American Heart Association guidelines and provides strong diagnostic evidence in some countries with national BCG vaccination policies, such as Japan, Korea, and Taiwan [2,7]. BCGitis appears in the early stage of disease; for this reason, infantile Kawasaki disease patients with BCGitis can be diagnosed and treated earlier, even if other diagnostic criteria are not fully present [20]. It also has a higher prevalence than erythema/edema of the extremities and cervical lymphadenopathy.

It has been reported that coronary artery lesions have a high occurrence rate of 30–80% in infantile Kawasaki disease [3,21,22,23]. This study also showed a similar trend to a previous study, in that the incidence of incomplete Kawasaki disease was high. However, there was a difference, in that the incidence of coronary artery lesions was lower than that of previous studies. It is thought that the occurrence of coronary artery lesions was low because early diagnosis and treatment were possible owing to the manifestation of BCGitis occurring in the early stage of the disease. The incidence of coronary artery lesions in Kawasaki disease with BCGitis was reported to be 10.3–34.1% [5,10,12,18]. An epidemiological study of Kawasaki disease in patients under three months of age in Korea reported no higher incidence of coronary artery lesions or coronary artery aneurysms compared with children older than three months of age [4].

According to the American Heart Association guidelines, careful observation of Kawasaki disease is important because not all clinical features appear simultaneously. A previous study reported three cases of patients who died without being diagnosed with Kawasaki disease due to a lack of clinical symptoms. The autopsy showed coronary lesions, such as large aneurysm of the coronary artery with branch obstruction by thrombi and total occlusion of the coronary artery, which might have been related to preceding Kawasaki disease. The authors emphasized the need for early treatment in young infants [24].

This study has demonstrated the importance of BCGitis in very young infants under three months of age.

A limitation of this study is the small number of patients aged less than three months among all the Kawasaki disease patients. Due to the small sample size, it was difficult to make statistical comparisons.

## 5. Conclusions

Kawasaki disease should be suspected in infants aged less than three months with unexplained fever and BCGitis, to assist the diagnosis of incomplete Kawasaki disease. BCGitis in infantile Kawasaki disease is a diagnostically useful clinical sign.

## Figures and Tables

**Figure 1 children-09-00857-f001:**
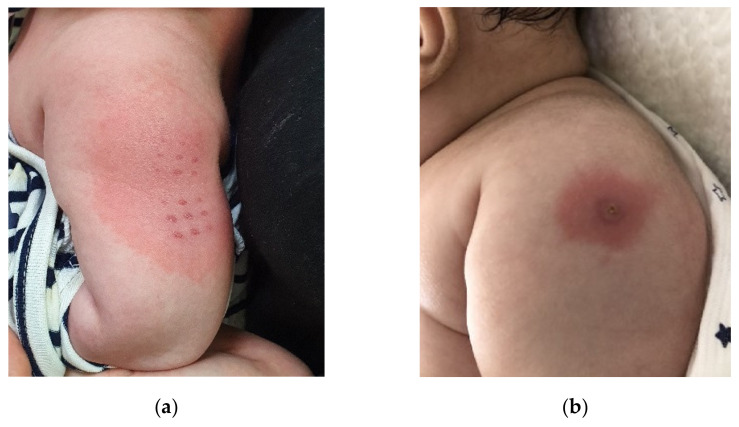
Erythema at the site of Bacille Calmette-Guerin (BCG) inoculation following two different delivery methods. (**a**) Transdermal method (**b**) Intradermal method.

**Table 1 children-09-00857-t001:** Comparison of demographic and clinical characteristics between groups.

	Group 1 (*n* = 19)	Group 2 (*n* = 454)	*p* Value
Number of patients (%)	19 (4%)	454 (96%)	
Age, months (median (IQR))	3 (2–2)	32 (15.0–48.7)	
Sex (male:female)	13:6	265:189	
Total duration of fever, days (median (IQR))	4 (3–4.5)	6 (5–7)	<0.001
Fever duration before IVIG administration, days (median (IQR))	2 (2–3)	5 (4–6)	<0.001
Incomplete Kawasaki disease	15 (79%)	267 (59%)	0.080
Clinical manifestations			
Erythema and cracking of lips & oral mucosa, strawberry tongue	14 (74%)	355 (78%)	0.642
Conjunctival injection	10 (53%)	360 (79%)	0.006
Skin rash	12 (63%)	318 (70%)	0.522
Erythema and edema of the hands and feet	8 (42%)	296 (65%)	0.040
Cervical lymphadenopathy	2 (11%)	256 (56%)	<0.001
Erythema and induration at BCG inoculation site	17 (89%)	145 (32%)	<0.001
Desquamation of fingers and toes	10 (53%)	106 (23%)	0.004
IVIG resistance	4 (21%)	162 (36%)	0.056
Coronary artery complication	2 (11%)	45(10%)	0.930
Dilation only (*Z* score 2 to <2.5)	0	16	
Small aneurysm (*Z* score ≥2.5 to <5)	1	22	
Medium aneurysm (*Z* score ≥5 to <10)	1	3	
Large and giant aneurysm (*Z* score ≥10)	0	4	

Values are presented as median (interquartile range) or absolute number (percentage). Abbreviations: IVIG, intravenous immunoglobulin; BCG, Bacille Calmette-Guerin.

**Table 2 children-09-00857-t002:** Comparison of laboratory findings of two groups on the first day of admission.

Laboratory Findings	Group 1 (*n* = 19)	Group 2 (*n* = 454)	*p* Value
WBC (10^3^/μL)	11.9 (9.7–15.6)	13.4 (10.3–17.0)	0.301
Hemoglobin (g/L)	101 (97–111)	117 (111–123)	0.187
Platelet (10^3^/μL)	436 (373–478)	353.5 (286.0–424.5)	0.031
CRP (mg/L)	0.3 (0.4–0.5)	0.5 (0.3–0.9)	0.049
ESR (mm/h)	31 (10–43)	54 (34–74)	0.002
Total protein (g/L)	58 (56–60)	67 (63–71)	<0.000
Albumin (g/L)	40 (36–40)	40 (38–43)	0.535
AST (U/L)	34 (25.5–42.5)	39 (29.0–75.7)	0.699
ALT (U/L)	29 (19.0–53.0)	24 (14.0–109.5)	0.321
Total bilirubin (μmol/L)	48.6 (39.4–67.2)	38.9 (26.5–59.2)	0.924
Sodium (mmol/L)	135 (133.5–136.0)	136 (134.0–138.0)	0.657
NT-proBNP (pg/mL)	873 (560–2311)	263 (93–1110)	0.034

Values are presented as median (interquartile range). Abbreviations: WBC, white blood cell; ESR, erythrocyte sedimentation rate; CRP, C-reactive protein; AST, aspartate transaminase; ALT, alanine transaminase; NT-proBNP, N-terminal pro-brain natriuretic peptide.

## Data Availability

Data from this study can be obtained by request to the authors.

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
