# Peer review of "Bacille Calmette-Guérin Site Reactivation of Kawasaki Disease in Infants under 3 Months of Age: Relation with Diagnosis and Prognosis"

_children, 2022, doi:10.3390/children9060857_

Round 1
Reviewer 1 Report
Authors tried to study the significance of BCG reactivation in <3 months old infants with KD. This is something new. Limitation is the very small number of infants in < 3 months - group1 compared to group 2.
1. Incidence of BCG'itis tends to be low in older children 10-12% compared to 40% in infants. This fact is clear in your study too within <3 mon old.
2. Diagnostic criteria revised in 2018 by AHA recommended checking CRP and ESR in children less than 1 year of age with fever >7 days in the presence of principal clinical features or with 2 or3 of the clinical features but with fever of >5days. Interestingly, AHA commended "An experienced clinician may even diagnose KD in children with only 3 days of fever" - (AHA 2018- Contemporary Pediatrics.2018,35,10.). This can be included in ur discussion.
3. About your Lab studies - ESR and NT-proBNP are statistically significant even though CRP not different. Even though not significant platelet count high in group 1 so early.
4. When making diagnosis of incomplete KD so early it is justifiable to rule out other exanthematous fevers which are more common in <3 months old. please refer to - Yim Ying Lim et al. Reactivation of BCG .............. BMJ Case Report 2020;13:e239648. - You can mention this in the discussion.
5. Comment: Did you find in your cohort any difference in the BCG reactivation in infants who received BCG by transdermal and intradermal methods. If so please make a note in results.
Author Response
Reviewer 1
Authors tried to study the significance of BCG reactivation in <3 months old infants with KD. This is something new. Limitation is the very small number of infants in < 3 months - group1 compared to group 2.
- Incidence of BCG'itis tends to be low in older children 10-12% compared to 40% in infants. This fact is clear in your study too within <3 mon old.
: I really appreciate to your opinion.
- Diagnostic criteria revised in 2018 by AHA recommended checking CRP and ESR in children less than 1 year of age with fever >7 days in the presence of principal clinical features or with 2 or3 of the clinical features but with fever of >5days. Interestingly, AHA commended "An experienced clinician may even diagnose KD in children with only 3 days of fever" - (AHA 2018- Contemporary Pediatrics.2018,35,10.). This can be included in ur discussion.
: I really appreciate to your opinion.
In page 6 (in discussion) “BCGitis are diagnostic criteria included in the Japanese Circulation Society / Japanese Society for Cardiovascular Surgery 2020 and American Heart Association guidelines and provide strong diagnostic evidence in some countries with national BCG vaccination policies, such as Japan, Korea, and Taiwan [2, 7]. BCGitis appears in the early stage of disease; for this reason, infantile Kawasaki disease patients with BCGitis can be diagnosed and treated earlier even if other diagnostic criteria are not fully present.”
I think this part has the same meaning as your comment.
- About your Lab studies - ESR and NT-proBNP are statistically significant even though CRP not different. Even though not significant platelet count high in group 1 so early.
: I really appreciate to your opinion. Although it is difficult to give a definite explanation for the increased platelet count from “very early” stage, I believe that it may be because the group 1 patients in our study had “highly active” inflammation. It is hoped that larger studies will be conducted in subsequent studies to elucidate the reason.
- When making diagnosis of incomplete KD so early it is justifiable to rule out other exanthematous fevers which are more common in <3 months old. please refer to - Yim Ying Lim et al. Reactivation of BCG .............. BMJ Case Report 2020;13:e239648. - You can mention this in the discussion.
: I really appreciate to your opinion. I mentioned that paper in the discussion and added to the reference list.
- Comment: Did you find in your cohort any difference in the BCG reactivation in infants who received BCG by transdermal and intradermal methods. If so please make a note in results.
: I really appreciate to your opinion. There was no significant difference between the two routes. The morphology of BCGitis can manifest different depending on the injection method, so I provide pictures of both methods, so I want even non-specialists can understand it well.

Reviewer 2 Report
Very well written and well conducted sudy. Considering the migratory movements of many children even from areas where BCG vaccination is mandatory to areas where it is not,I think it is important to underline the value of a BCGite among the possible diagnostic clues for KD, especially in those patients yonger than 3 months of age, for which the diagnosis is more difficult,
and in practice it can be done in late and expose to more complications. The study also considers, the relationship between BCGite, even if other principal criteria for KD were insufficient, and changes in the clinical course and cardiac complications
Author Response
Very well written and well conducted sudy. Considering the migratory movements of many children even from areas where BCG vaccination is mandatory to areas where it is not,
I think it is important to underline the value of a BCGite among the possible diagnostic clues for KD, especially in those patients yonger than 3 months of age, for which the diagnosis is more difficult,
and in practice it can be done in late and expose to more complications. The study also considers, the relationship between BCGite, even if other principal criteria for KD were insufficient, and changes in the clinical course and cardiac complications
: I really appreciate to your opinion.